# PHASE: PHysically-grounded Abstract Social Events for Machine Social Perception

**Aviv Netanyahu**[*]    **Tianmin Shu**[*]    **Boris Katz**    **Andrei Barbu**    **Joshua B. Tenenbaum**

{avivn, tshu, boris, abarbu, jbt}@mit.edu
Massachusetts Institute of Technology, Cambridge, MA 02139

## Abstract

The ability to perceive and reason about social interactions in the context of physical environments is core to human social intelligence and human-machine cooperation. However, no prior dataset or benchmark has systematically evaluated physically grounded perception of complex social interactions that go beyond short actions, such as high-fiving, or simple group activities, such as gathering. In this work, we create a dataset of physically-grounded abstract social events, PHASE, that resemble a wide range of real-life social interactions by including social concepts such as helping another agent. PHASE consists of 2D animations of pairs of agents moving in a continuous space generated procedurally using a physics engine and a hierarchical planner. Agents have a limited field of view, and can interact with multiple objects, in an environment that has multiple landmarks and obstacles. Using PHASE, we design a social recognition task and a social prediction task. PHASE is validated with human experiments demonstrating that humans perceive rich interactions in the social events, and that the simulated agents behave similarly to humans. As a baseline model, we introduce a Bayesian inverse planning approach, SIMPLE (SIMulation, Planning and Local Estimation), which outperforms state-of-the-art feed-forward neural networks. We hope that PHASE can serve as a difficult new challenge for developing new models that can recognize complex social interactions.[1]

## 1   Introduction

Humans make spontaneous and robust judgments of others' mental states (e.g., goals, beliefs, and desires), characteristics (e.g., physical strength), and relationships (e.g., friend, opponent) by watching how other agents interact with the physical world and with each other. These judgements are critical to engaging socially with other agents. AI and robots that cooperate with humans will similarly need to engage with us socially, and by extension make these same judgements about both physical notions, like strength, and social notions, like mental states.

Prior work has looked at recognizing social interactions, but evaluations and benchmarks have been limited to the artifacts of social interactions, like high-fives and hand shakes [18, 14, 17, 24, 7, 22, 19, 12, 6, 15, 25]. Here, we create the first benchmark for reasoning about the underlying mechanisms and beliefs of social interactions, instead of these overt actions that are correlated with certain types of interactions. We propose a novel dataset, PHASE (PHysically-grounded Abstract Social Events), that expresses complex social concepts in a physical setting, such as helping and hindering.

---

[*]These two authors contributed equally.
[1]Project website: https://www.tshu.io/PHASE.

2nd Workshop on Shared Visual Representations in Human and Machine Intelligence (SVRHM), NeurIPS 2020.

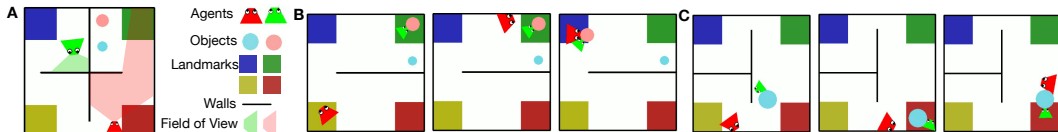

Figure 1: Demonstrating the PHASE physical-social simulation. (**A**) The elements of the simulation: agents (with a limited conical field of view), objects with different colors and sizes, landmarks with different colors, and immovable walls. (**B**) Frames from a video depicting an abstract social event sampled in the PHASE simulator. The green agent is weak, therefore has difficulty moving the pink object, which the red agent eventually helps with. (**C**) Frames from a video depicting the opposite situation, where a green agent is moving an object when the red agent steps in and steals it away.

Collecting datasets of social interactions is very difficult, as it involves playing out complex scenarios while recording the intent and mental states of agents. Heider and Simmel [8] demonstrated that one can understand social events depicted in animations of simple geometric shapes moving in a physical environment. We draw inspiration from this to propose a joint physical-social simulation for generating PHASE, as shown in Figure 1, where agents and objects are physical bodies moving in a 2D physics simulation. Grounded social interactions are generated using a hierarchical planner and a physics engine (Figure 2). Manipulating the parameters of this simulation enables us (i) to procedurally generate complex social events that resemble a wide range of real-life social interactions as training data for models, and (ii) to control social and physical variables to create training data with balanced ground-truth labels as well as to carefully design evaluation for generalization in unseen environments and social behaviors.

We propose two machine social perception tasks on PHASE that require recognizing goals and relationships of agents, and predicting the future trajectories of agents. We test state-of-the-art methods based on feed-forward neural networks and show that they fail to understand or predict many of these social interactions. To further augment machine perception of social interactions, we introduce a Bayesian inverse planning-based approach, SIMPLE (SIMulation, Planning and Local Estimation), that significantly outperforms prior work.

In summary, our contributions include: (i) a joint physical-social simulation for procedurally generating abstract social events grounded in physical environments, (ii) using this engine to generate a first-of-its-kind abstract social events dataset, and (iii) proposing two social perception tasks and a benchmark including state-of-the-art methods and a Bayesian inverse planning-based approach.

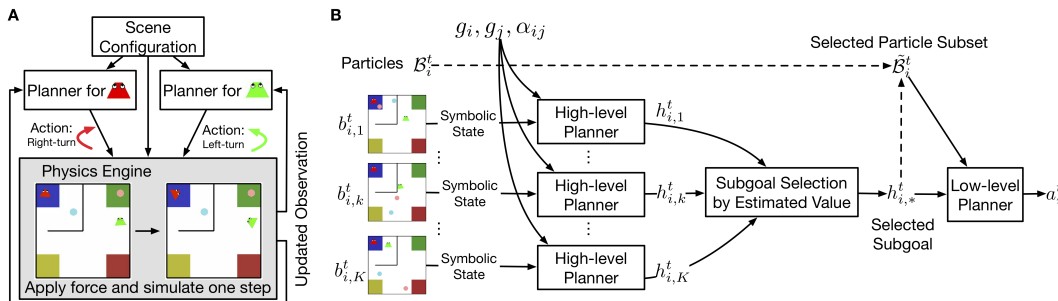

Figure 2: Overview of the simulation and the hierarchical planner. (**A**) Key components of the simulation. (**B**) The hierarchical planner in our simulation. At each step the planner searches for an action based on the agent's belief represented by a set of particles.

## 2 Joint Physical-Social Simulation

The simulation objective is to synthesize motion trajectories of multiple entities (agents and objects) that follow physical dynamics, and also elicit strong impression of social behaviors. As shown in Figure 2A, the simulation has three main components — a structured physical and social scene configuration, a hierarchical planner, and a physics engine. To synthesize an abstract social event, we specify the physical and social configurations. Each agent has an independent hierarchical planner

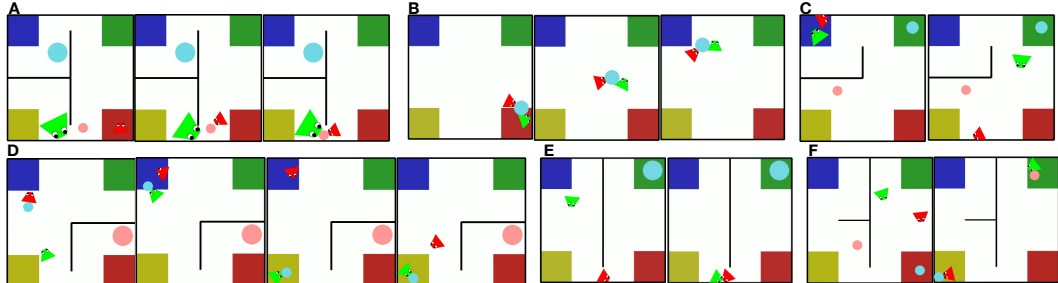

Figure 3: Example abstract social events in the PHASE dataset. (**A**) Helping a large-sized agent get an object it could not reach. (**B**) Two weak agents carrying an object together (collaboration). (**C**) Green chasing red. (**D**) Two agents trying to put the same object to different landmarks. (**E**) Red blocking green (hindering). (**F**) Neutral agents pursing independent goals.

that has access only to its own mental state and partial view of the environment. At each step, each agent replans based on its beliefs and observations of the scene. The physics engine, then steps the environment forward based on these plans. This process repeats to generate a video. We give an overview of the simulation in this section and provide more details in Appendix B.

## 2.1 Formulation

We formally define the social behaviors of agents by a decentralized partially observable Markov decision process (Dec-POMDP) [16]. There are $N$ agents sharing the same state space $\mathcal{S}$ and action space $\mathcal{A}$. In our simulation, the action space consists of applying a force in one of 8 directions, turning right or left, stopping, grabbing an object (attaching it to the agent's body) or letting go of an object, and no force. The physical dynamics of the environment is defined by state transition probabilities $\mathcal{T} : \mathcal{S} \times \mathcal{A}^N \times \mathbb{R}^N \rightarrow \mathcal{S}$, i.e., $P(s'|s, \{a_i\}_{i=1}^N, \{f_i\}_{i=1}^N)$, where $f_i \in \mathbb{R}$ is the maximum magnitude of the force agent $i$ can exert at one step, defining the agent's physical strength.

At each step $t$, agent $i$ observes part of the world state $s^t$ through vision (which is limited to a conical field of view obstructed by walls and entities) and a touch sensor, i.e., $o_i^t \sim O_i(o|s^t)$. The agent updates its belief, $b(s^t)$, based on the observation by $b(s^{t+1}) \propto O_i(o|s^{t+1}) \sum_{s^t \in \mathcal{S}} P(s^{t+1}|s^t, \{a_i\}_{i=1}^N, \{f_i\}_{i=1}^N) b(s^t)$. Agents know the underlying map of the environment and the number of entities, but not where other entities are unless they are seen or felt.

Each agent has a physical goal $g_i \in \mathcal{G}$ or a social goal, i.e., helping or hindering. Social goals are indicated by a social utility weight $\alpha_{ij} \in \{-1, 0, 1\}$. When $\alpha_{ij} = 1$, agent $i$ will help agent $j$ achieve its goal; when $\alpha_{ij} = -1$, agent $i$ will hinder agent $j$; when $\alpha_{ij} = 0$, agent $i$ will pursue its own physical goal. According to this, we can write an agent's reward in the context of a 2-agent interaction: $R_i(s, a) = (1 - |\alpha_{ij}|)R(s, g_i) + \alpha_{ij}R(s, g_j) + C(a)$, where $C(a)$ is the cost of taking action $a$. Given this reward function, each agent plans its actions to maximize accumulated reward over a limited horizon $T$, i.e., $\sum_{t=1}^T R_i(s^t, a_i^t)$. We assume agents know each other's goals, which is shown to be sufficient for generating rich social behaviors in our experiments.

## 2.2 Hierarchical Planner

It is challenging to synthesize complex social behaviors at scale. Prior work attempted to do this with manually designed motion [4]. In this work, we propose a hierarchical planner as shown in Figure 2B for deriving agent behaviors with bounded rationality, which is inspired by task and motion planning (TAMP) [11]. The planner maintains a set of particles to approximate the belief of each agent at each step. At each step, we first update particles by simulating one step in the physics engine and then resample the particles that violate the new observation. Given the current particles, we use a high-level planner to generate subgoals that are represented by predicates indicating which immediate states an agent should reach in order to achieve the final goal. The high-level planner will select the most valuable subgoal at the current step, favoring a subgoal that frequently appears in the subgoal plans among the particles, and has a lower cost. Finally, we feed the subset of the particles that yield the best subgoal to the low-level planner, which will search for the best action to reach that subgoal.

# 3 PHASE Dataset

## 3.1 Procedural Generation

To synthesize the PHASE dataset, we sample a rich set of scene configurations variables, each of which is fed to the simulation to render a video depicting an abstract social event. **Physical Variables:** There are 90 different environment layouts, comprised of wall positions and sizes. There are four possible sizes for entities and four agent strength levels. There two agents, and up to two objects. **Social Variables:** We sample either a physical goal or a social goal for each agent. The physical goals are: going to one of the four landmarks, moving a specific object to one of the four landmarks, approaching another agent, and getting away from another agent. As there could be two different objects, we have 14 physical goals in total. There are two social goals — helping and hindering.

By sampling the environment layout, entity sizes, agent strengths, agent goals, $\alpha_{ij}$ and $\alpha_{ji}$, and the initial states of all entities, we can create a large set of physical and social scene configurations. In general, there are five types of social events: (i) helping, (ii) collaborating on a joint goal, (iii) hindering, (iv) two agents having conflicting goals such as chasing or fighting over the same object, and (v) two neutral agents pursing independent goals. We show examples of these social events in Figure 3 and in the supplementary video. Finally, we define three types of relationships between agents based on these five types of social events: (i) and (ii) correspond to friendly relations, (iii) and (iv) correspond to adversarial relations, and (v) corresponds to neutral relations.

**Dataset Statistics.** PHASE contains 500 videos of abstract social events. Each lasts from 10 seconds to 25 seconds. Each goal has 37 to 65 examples. For the friendly, adversarial, and neutral relations, there are 181, 195, and 124 examples respectively. With these 500 videos, we create a training set of 320 videos, a validation set of 80 videos, and a testing set of 100 videos. To evaluate the generalization of a trained model, 80% of the testing videos are synthesized with novel environment layouts that are unseen in the training and validation sets. Moreover, there are 10 videos showing unique types of social interactions that are only seen in the test set.

## 3.2 Human Experiments

To evaluate whether PHASE depicts social interactions, we conduct two human experiments on Mechanical Turk.

**Experiment 1: Multi-label descriptions.** We compiled a set of 23 social interaction types from (i) common social interactions studied in prior literature [4, 5], and (ii) free responses collected from a preliminary experiment where participants described videos in PHASE in their own words. We found that the abstract social events in PHASE resemble 18 diverse real-life interaction categories (Figure 4), participants could recognize unintentional interactions (e.g., agents with independent goals accidentally crossed paths), and all friendly and adversarial interactions were meaningful and intentional to participants.

**Experiment 2: Comparing the synthesized trajectories with human-controlled trajectories.** This experiment consists of two parts. In the first part, we designed a 2-player game based on PHASE, where humans can control agents by pressing keys. In the second part, we recruited additional participants and divided them into two groups. One group watched the human-controlled videos, and the other watched matching videos from PHASE. For each video, participants were asked to judge the goals and

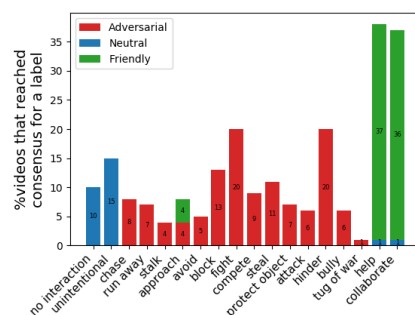

Figure 4: Consistent human responses in Experiment 2 showing how many videos (percentages) were assigned with an interaction category by at least 50% of participants who watched the videos.

relations of the agents, and rate how likely humans were to behave similarly to these agents under the same goals and relations. We found that there was no significant difference between the two groups ($t(15) = -0.96, p = 0.34$ for goal judgments, $t(7) = -1.5, p = 0.17$ for relation judgments, and $t(7) = -0.21, p = 0.83$ for human-likelihood).

| Method | Goal | Relation |
|---|---|---|
| Human | 0.975 | 0.97 |
| 2-Level LSTM | 0.370 | 0.73 |
| ARG | 0.420 | 0.75 |
| SIMPLE | 0.870 | 0.88 |

Table 1: Goal and relation recognition accuracy in the first task.

| Method | ADE | | | FDE | | |
|---|---|---|---|---|---|---|
| | 25% | 50% | 75% | 25% | 50% | 75% |
| S-LSTM | 6.19 | 6.58 | 6.75 | 6.27 | 6.58 | 6.74 |
| STGAT | 6.01 | 6.40 | 6.51 | 6.06 | 6.30 | 6.44 |
| SIMPLE | 2.78 | 1.93 | 1.72 | 2.63 | 2.28 | 1.75 |

Table 2: Trajectory prediction error for different portions of the videos in the second task.

### 3.3 Social Perception Tasks

We design two social perception tasks that evaluate a model's abilities to recognize the goals and relations of agents, and to predict the future social behaviors based on partial observation.

**Task 1: Joint inference of goals and relations.** This task focuses on understanding social interactions, i.e., jointly inferring agents' goals and relationships to other agents to explain their behavior. Unlike typical activity recognition, this task focuses on *why* the agents exhibit certain behaviors, rather than giving a literal description of *what* the agents are doing.

**Task 2: Multi-entity trajectory prediction.** Since robots and intelligent machines must not only understand social interactions, but also engage with us socially, we design a second task to predict the behavior of a social agent. This requires both social and physical reasoning, as all agents and objects are constrained by physics. A model must predict the trajectories in the next 10 steps (2.5 seconds) of all entities after watching the first 25%, 50%, or 75% of the video.

## 4 Results

To address the proposed tasks, we develop a Bayesian inverse planning approach, SIMPLE , that integrates computational theory of mind [2] with simulation for physical reasoning [3] (see Appendix D for the details of this approach). For the first task, joint goal and relation inference, we compare SIMPLE with two state-of-the-art approaches for recognizing group activities, 2-Level LSTM [10] and ARG [23], as well as with human performance. For the second task, trajectory prediction, we compare SIMPLE with two feed-forward models: Social-LSTM [1] and STGAT [9]. We use ADE and FDE [1] as evaluation metrics. We present more details of the baselines in Appendix F.

Table 1 and Table 2 summarize the performance of all methods in the two tasks. For the first task, humans achieve almost perfect accuracy. SIMPLE performs significantly better than the other two baselines. This suggests that the underlying meaning of different social interactions could not be captured by motion patterns alone. Similarly, trajectories prediction based on SIMPLE also outperforms Social-LSTM and STGAT that have been demonstrated to be effective in predicting human pedestrian trajectories. Crucially, the predicted trajectories from these two models deviate further from the ground-truth as the observations accumulate, whereas predictions based on inferences from SIMPLE are increasingly accurate due to better goal inference when observing longer trajectories.

Although SIMPLE demonstrates superior results compared to strong baselines, it requires simulation in a physics engine and expensive search with a planner, which poses challenges for future work on machine social perception. E.g., how to achieve success in theory-based inference and social behavior prediction for complex physically grounded social interactions? How can models generalize social and physical dynamics learned from training environments to novel environments?

## 5 Conclusion

We propose a joint physical-social simulation to procedurally generate a large set of social interactions grounded in physical environments. We use this simulator to create the first physically-grounded abstract social event dataset, PHASE. Our human experiments show that the synthesized videos are recognized as depicting a large variety of real-life social interactions. The two social perception tasks for machines demonstrate that much remains to be done with existing models, even with the Bayesian inverse-planning approach, SIMPLE, we introduce for solving these tasks. Having a systematic benchmark for understanding social interactions will hopefully spur new research and models.

## Acknowledgments

We would like to thank David Mayo and Yen-Ling Kuo for their help with the experiments. This work was supported by NSF STC award CCF-1231216 (the Center for Brains, Minds and Machines), ONR MURI N00014-13-1-0333, the MIT-Air Force AI Accelerator, Toyota Research Institute, the DARPA GAILA program, and the ONR Science of Artificial Intelligence program.

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

| Method | Goal | Relation |
|---|---|---|
| Human | 1.0 | 1.0 |
| 2-Level LSTM | 0.063 | 0.38 |
| ARG | 0.25 | 0.25 |
| SIMPLE | 0.875 | 1.0 |

Table 3: Goal and relation recognition accuracy evaluated on *human-controlled* trajectories.

| Method | ADE | | | FDE | | |
|---|---|---|---|---|---|---|
| | 25% | 50% | 75% | 25% | 50% | 75% |
| S-LSTM | 6.13 | 6.10 | 6.12 | 6.17 | 6.13 | 6.23 |
| STGAT | 6.07 | 6.04 | 6.97 | 6.09 | 6.05 | 6.13 |
| SIMPLE | 3.60 | 3.01 | 1.97 | 3.32 | 2.64 | 1.17 |

Table 4: Trajectory prediction accuracy after watching different portions of the videos evaluated on *human-controlled* trajectories.

| Method | Goal | Relation |
|---|---|---|
| SIMPLE (Initial) | 0.795 | 0.82 |
| SIMPLE (Global) | 0.835 | 0.84 |
| SIMPLE (Full) | 0.870 | 0.88 |

Table 5: Ablation study on the effect of proposal updates based on local estimation for Task 1. The evaluation is using the test set of PHASE. Initial, Global, and Full represent SIMPLE with only the initial proposals, SIMPLE using global estimation for updating the proposals, and SIMPLE as proposed in Algorithm 2.

# A    Additional Experimental Results

## A.1    Evaluation on Human-controlled Trajectories

We evaluate the baseline approaches on the two tasks using the human-controlled trajectories (8 videos) collected in the second human experiment (comparing the synthesized trajectories with human-controlled trajectories) as summarized in Table 3 and Table 4. The results show that the approaches based on feed-forward neural nets suffer a high degradation in the recognition accuracy for goals and relations for Task 1, whereas the performance of SIMPLE is consistent with the results on synthesized trajectories in PHASE. For Task 2, the prediction error by SIMPLE is initially higher than the error when using PHASE; but it gradually decreases as SIMPLE observes a larger portion of the trajectories, and eventually becomes comparable to the error on PHASE.

## A.2    Ablation Study (Evaluation on PHASE)

To evaluate the effect of the proposal update based on location estimation in SIMPLE, we compare the full approach with (i) a variant without proposal update, and (ii) a variant with update based on the complete trajectories instead of location estimation (also with 6 iterations). We report the performance for Task 1 on the test set of PHASE in Table 5. The results demonstrate that local estimation indeed can help find better proposals through a handful of iterations.

# B    Details of Joint Physical-Social Simulation

Our joint physical-social simulation is described in Algorithm 1, which includes a physical simulation $\mathcal{T}$, and a hierarchical planner which consists of a high-level planner (HP) and a low-level planner (LP). Given the scene configuration, the simulation updates the belief particles based on new observations, uses the hierarchical planner to sample actions for all agents based on the updated particles, feeds the actions to the physics engine to simulate one step, and renders 5 frames of video based on the simulated physical states. The final video has a frame rate of 20 FPS. We discuss more implementation details as follows.

**Algorithm 1:** Joint Physical-Social Simulation

---

**Input:** $g_1, g_2, \alpha_{12}, \alpha_{21}, f_1, f_2$, and initial state $s^1$
**Output:** Abstract social event $s^{1:T}$
**for** *agent* $i = 1, \cdots, 2$ **do**
$\quad$ Initialize belief particles $\{b^0_{i,k}\}^K_{k=1}$ ;
**end**
**for** *time steps* $t = 1, \cdots, T$ **do**
$\quad$ **for** *agent* $i = 1, \cdots, 2$ **do**
$\quad\quad$ Update observation $o^t_i$;
$\quad\quad$ Update belief particles $\{b^t_{i,k}\}^K_{k=1}$ based on $o^t_i$;
$\quad\quad$ Set the other agent $j \leftarrow \{1, 2\} \setminus \{i\}$;
$\quad\quad$ **for** *each particle* $k = 1, \cdots, K$ **do**
$\quad\quad\quad$ Get subgoal $h^t_{i,k} \leftarrow \text{HP}(g_i, g_j, \alpha_{ij}, b^t_{i,k})$;
$\quad\quad$ **end**
$\quad\quad$ **for** *subgoal* $h \in \mathcal{H}$ **do**
$\quad\quad\quad$ Esitmate value $V(\mathcal{B}^t_i, h, g_i, g_j, \alpha_{ij}) =$
$\quad\quad\quad$ $\frac{1}{K} \sum^K_{k=1} \mathbb{1}(h = h^t_{i,k}) - \frac{\lambda}{\sum^K_{k=1} \mathbb{1}(h=h^t_{i,k})} \sum^K_{k=1} \mathbb{1}(h = h^t_{i,k}) \hat{C}(b^t_{i,k}, s_g)$;
$\quad\quad$ **end**
$\quad\quad$ Select subgoal $h^t_{i,*} = \arg\max_h V(\mathcal{B}^t_i, h, g_i, g_j, \alpha_{ij})$;
$\quad\quad$ Get belief particles $\tilde{\mathcal{B}}^t_i$ that correspond to $h^t_{i,*}$;
$\quad\quad$ Get action $a^t_i \leftarrow \text{LP}(\tilde{\mathcal{B}}^t_i, h^t_{i,*})$;
$\quad$ **end**
$\quad$ Update state $s^{t+1} \leftarrow \mathcal{T}(s^t, \{a^t_i\}^2_{i=1}, \{f_i\}^2_{i=1})$;
**end**

---

| Predicate | Definition |
|---|---|
| ON(*agent/object*, *landmark*) | An entity is on a landmark |
| TOUCH(*agent*, *agent/object*) | An agent touches another entity |
| ATTACH(*agent*, *object*) | An object is attached to an agent's body |
| CLOSE(*agent/object*, *agent/object/landmark*) | An entity is within a certain distance away from another entity or a landmark |

Table 6: Predicates and their definitions. Note that we also consider their negations, which are not shown in the table for brevity.

## B.1 Predicates, Symbolic States, Goals, and Subgoals

In our simulation, we define a set of predicates as summarized in Table 6. These predicates and their negations are used to (i) convert a physical state into a symbolic state, and also (ii) become a subgoal space that our hierarchical planner considers for the high-level plans.

Furthermore, the final goal states for physical goals and social goals of agents are also represented by a subset of these predicates, i.e., ON(*agent/object*, *landmark*), TOUCH(*agent*, *agent)*, and their negations.

## B.2 Hierarchical Planner

The planner maintains a set of particles to approximate the belief of each agent at each step, i.e., $\mathcal{B}^t_i = \{b^t_{i,k}\}^K_{k=1}$, where each particle $b^t_{i,k}$ represents a possible world state. All particles are initially sampled from a uniform distribution of possible states of entities. At each step, we first update particles by simulating one step in the physics engine assuming that other agents will maintain a constant motion and then resample the particles that violate the new observation.

Given the current particle set, we use a high-level planner to generate subgoals. The high-level planner first converts the physical state in each particle into symbolic states represented by predicates, and then searches for the best symbolic plan based on the reward of each agent. Subgoals

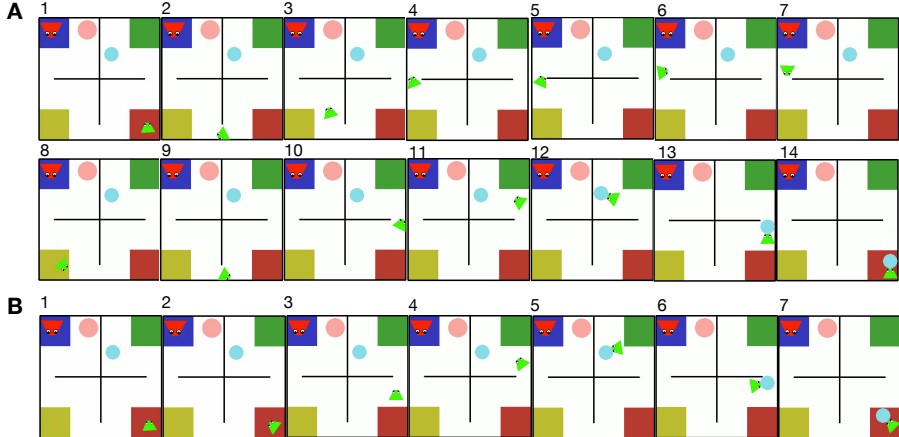

Figure 5: Illustration of the effect of estimated value function for the high-level planner. The numbers indicate temporal order of the frames. In both sequences, the green agent's goal is to move the blue object to the red landmark in the bottom-right corner. Since it does not see the blue object initially, it needs to first find the object. (**A**) The sequence when $\lambda = 0$. since there is more unseen space in the left part of the environment, it is more likely that the blue object is in the left part. So the green agent first searches the left part when not considering the cost of doing so. (**B**) The sequence when $\lambda = 0.05$. When considering the cost, it is worthwhile for the green agent to search the nearby region first. The chance of finding the blue object there is slightly lower than the left region, but the resulting cost is considerably lower. In particular, it first looks around (frame #2) and then proceeds to search the upper-right part (frame #3 and #4). This comparison demonstrates that an appropriate $\lambda$ could give us more natural agent behaviors under partial observability.

are represented by predicates indicating which immediate states an agent should reach in order to achieve the final goal. This produces a subgoal space $\mathcal{H}$ consisting of all possible predicates. For computational efficiency, we only consider the most immediate subgoal in the plan for the next move. Let $h_{i,k}^t \in \mathcal{H}$ be the best subgoal for agent $i$ at step $t$ based on its belief state in particle $b_{i,k}^t$. We estimate the value of each subgoal by $V(\mathcal{B}_i^t, h, g_i, g_j, \alpha_{ij}) = 1/K \sum_{k=1}^K \mathbb{1}(h = h_{i,k}^t) - \lambda/(\sum_{k=1}^K \mathbb{1}(h = h_{i,k}^t)) \sum_{k=1}^K \mathbb{1}(h = h_{i,k}^t) \hat{C}(b_{i,k}^t, s_g)$, where $\hat{C}(b_{i,k}^t, s_g))$ is a heuristics-based estimation of cost to reach goal state $s_g$ based on belief state $b_{i,k}^t$ defined as the estimated distance that the agent needs to travel before reaching the final goal state, and $\lambda \in (0, 1)$ is a scaling factor. The high-level planner will select the most valuable subgoal at the current step, i.e., $h_{i,*}^t = \arg\max_h V(\mathcal{B}_i^t, h, g_i, g_j, \alpha_{ij})$. Finally, we feed the subset of the particles that yield $h_{i,*}^t$ as the best subgoal ($\tilde{\mathcal{B}}_i^t \subset \mathcal{B}_i^t$) to the low-level planner, which will search for the best action to reach that subgoal.

For the **high-level** planner we use A* to search for a plan of subgoals for $N = 2$ agents based on $K = 50$ belief particles.

To ensure a subgoal selection for simulating natural agent behavior without expensive computation, we design a heuristics-based value estimation $V(\mathcal{B}_i^t, h, g_i, g_j, \alpha_{ij})$ for each subgoal as shown in Algorithm 1. This value function favors subgoals that are more likely to be the best subgoal in the true state (i.e., high frequency subgoals generated by all belief particles) and have lower cost (i.e., $\hat{C}$ estimated by the distance from the current state to the final goal state according to a given belief particle). By changing the weight $\lambda$, we are able to alter the agent's behavior. Figure 5 demonstrates an example of how $\lambda$ affects the agent's behavior. In practice, we find $\lambda = 0.05$ offers a good balance and can consistently generate natural behaviors.

For the **low-level** action planner, we use POMCP [21] with 1000 simulations and 10 rollout steps. For exploration in POMCP, we adopt a variant of PUCT algorithm introduced in [20], where we use $c_{\text{init}} = 1.25$ and $c_{\text{base}} = 1000$.

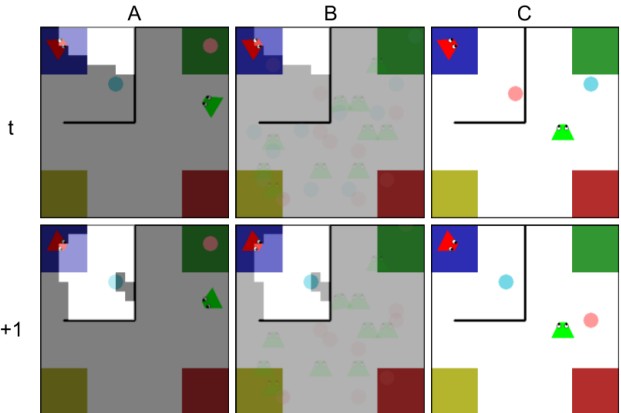

Figure 6: Illustration of how the red agent updates its belief using $K = 10$ particles. (**A**) True states $s^t$ (top) and $s^{t+1}$ (bottom). The bright pixels indicate the red agent's field of view. (**B**) $\{b^t_{\text{red},k}\}^K_{k=1}$ (top) and $\{b^{t+1}_{\text{red},k}\}^K_{k=1}$ (bottom). The states in the all particles are visualized together. At step $t + 1$ the red agent observes the blue object via its field of view. All particles are then updated accordingly with the ground truth properties of the blue object, and the inconsistent belief states are also resampled. (**C**) The state in one of the belief particles, $b^t_{\text{red},k}$ (top) and $b^{t+1}_{\text{red},k}$ (bottom). The particle is updated with ground truth properties blue object at step $t + 1$. The properties of the pink object are resampled at step $t + 1$ since its believed position in step $t$ conflicts with the observation at step $t + 1$.

### B.3 Belief Representation and Update

Each agent's belief is represented by $K = 50$ particles in the simulation. Each particle represents a possible world state that is consistent with the observations. The state in a particle includes the environment layout, and physical properties of each entity — shape, size, center position, orientation of the body, linear and angular velocity, and attached entities.

Each particle is updated with the ground truth properties of *observed* entities: the agent itself, other entities in its field of view (approximated by $1 \times 1$ grid cells on the map) or entities in contact with the agent. Entities that are in contact with observed entities are also defined as observed.[2] Contact occurs when entities are attached or collide, and is signaled by agents' touch sensory.

Unobserved entity properties differ between particles. We start by randomly sampling possible initial positions from the 2D environment and setting other properties (orientation and velocity) to 0. To update a belief particle from $t$ to $t + 1$, we first apply the physics engine to simulate one step, where we assume constant motion for entities. Then we check the consistency between the simulated state at $t + 1$ and the actual observation at $t + 1$. For entities that contradict the observation, we resample their positions and orientations. We then repeat the consistency check and resampling until there is no conflict.

Figure 6 depicts an example of how an agent updates its belief from step $t$ to step $t + 1$ based on its observation at step $t + 1$.

## C  More Example Events in PHASE

We show more example events in PHASE in the supplementary video.

## D  Details of SIMPLE

To address the proposed tasks, we develop a Bayesian inverse planning approach, SIMPLE (SIMulation, Planning and Local Estimation), that integrates computational theory of mind [2] with simulation for physical reasoning [3].

---

[2]This is to ensure that the agent knows (i) whether there is *any* other agent grabbing the same object it is currently grabbing, and (ii) whether an observed agent is grabbing *any* object.

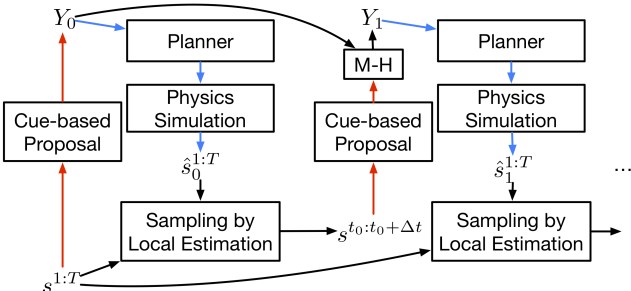

Figure 7: Diagram of how the proposal in a single particle is updated in SIMPLE. For brevity, we drop the subscript $m$. M-H represents Metropolis–Hastings algorithm for determining whether to accept the new proposal. The red lines indicate bottom-up proposals and the blue lines represent top-down generative processes.

Inverse planning relies on the notion that if one correctly infers hidden variables, like goals, relations, strengths, and beliefs, the generated rational plans for agents will be a good match for the observed plans. We instantiate this idea in the following way: let $Y = \langle g_i, g_j, \alpha_{ij}, \alpha_{ji}, f_i, f_j \rangle$ be the hypothesis, $s^{1:T}$ be the observed state sequence (in particular, trajectories of all entities) of the event, and $\hat{s}^{1:T} = G(g_i, g_j, \alpha_{ij}, \alpha_{ji}, f_i, f_j)$ be the simulation given the hypothesis, where the generative model $G(\cdot)$ includes both the hierarchical planner and the physics engine. Then we have the following posterior probability for inference

$$
\begin{aligned}
&P(Y = \langle g_i, g_j, \alpha_{ij}, \alpha_{ji}, f_i, f_j \rangle | s^{1:T}) \\
&\propto P(s^{1:T}|Y)P(g_i)P(g_j)P(\alpha_{ij}, \alpha_{ji})P(f_i)P(f_j),
\end{aligned}
\tag{1}
$$

where $P(s^{1:T}|Y) = e^{-\beta \sum_{t=1}^{T} ||s^t - \hat{s}^t||_2}$ is the likelihood based on the distance between the observed trajectories and the simulated trajectories w.r.t. the hypothesis, and $\beta > 0$ is a constant coefficient.

To efficiently explore the large hypothesis space, we perform probabilistic inference based on data-driven Markov Chain Monte Carlo (MCMC) that utilizes both cue-based bottom up proposals and top-down generative processes, as shown by Figure 7. We outline inference in two main steps as follows and discuss additional implementation details below.

**Initial Proposals.** Even though visual cues of trajectories alone may not give us the most accurate inference, they can provide reasonable guesses which may shrink the search space, thereby increasing the chance of making good proposals. Thus we use a bottom-up proposal approach that estimates the likelihood of pursuing a goal by the distance between the final state and the goal state as well as the change in that distance compared to the start of the video. For the social utility weights, $\alpha$, we adopt a uniform distribution. For strengths, we train a regression model based on a 2-layer MLP which takes in the average, maximum, and minimum velocities as well as accelerations of each agent. We describe the proposal approach in details in Appendix D.1 We sample $M$ particles to approximate the true posterior probability (Eq. 1), each of which contains an initial proposal $Y_{0,m}$ sampled from a cue-based proposal distribution, $Q(Y|s^{1:T})$.

**Proposal Update based on Local Estimation.** We run multiple iterations to update the proposals. Given the proposals at iteration $l$, we simulate the trajectories, i.e., $\hat{s}_{l,m}^{1:T}$, $\forall m = 1, \cdots, M$, and compare them with the observed trajectories, $s^{1:T}$. For each proposal, we sample a time interval with a fixed length, $\Delta T$, based on the errors between the simulation and the observations, i.e., $t_{l,m} \propto e^{\eta \sum_{\tau=t_{l,m}}^{t_{l,m}+\Delta T} ||\hat{s}_{l,m}^{\tau} - s^{\tau}||_2}$, where $\eta = 0.1$. The intuition behind this is that local deviation is often more informative in terms of how the proposal should be updated compared to the overall deviation.[3] After selecting a local time interval, we use the same bottom-up mechanism to again propose a new hypothesis for each particle, $Y_m'$, based only on $S' = s^{t_{l,m}:t_{l,m}+\Delta T}$. We then use the Metropolis–Hastings algorithm to decide whether to accept this new proposal for the particle, where the acceptance rate is $\alpha = \min\{1, \frac{Q(Y'|S')P(s^{1:T}|Y')}{Q(Y_{l,m}|S')P(s^{1:T}|Y_{l,m})}\}$.

---

[3]E.g., in hindering interactions, it is often not clear which physical goal was being hindered once two agents made contact; however, the first part of the video may reveal more information about what an agent's physical goal was since the agent was pursing that goal without interference from the other agent who was far away.

When planning the actions at step $t$, the planner utilizes the belief inferred from agents' past observation upon $t$. We achieve this by estimating the observations of agents at each step using the simulator, and then sample belief particles for each agent that are consistent with what that agent has seen. This purely bottom-up belief estimation can adequately approximate the true beliefs of agents while being computationally efficient. In contrast, proposing beliefs top-down would be intractable due to the large state space.

To approximate the posterior probability, we compute the weight for each particle $m$ at iteration $l$ as $w_{l,m} = P(s^{1:T}|Y_{l,m})/\sum_{k=1}^{M} P(s^{1:T}|Y_{l,k})$. Then an agent's goal can be inferred by

$$P(g_i|s^{1:T}) = \sum_{m=1}^{M} \mathbb{1}(g_i \in Y_{l,m})w_{l,m}, \tag{2}$$

where $\mathbb{1}(g_i \in Y_{l,m})$ indicates whether $g_i$ appears in the hypothesis $Y_{l,m}$. Similarly, we can compute $P(\alpha_{ij}|s^{1:T})$ and $P(\alpha_{ji}|s^{1:T})$.

Based on the final proposals and their weights, we can define posterior probability of the relationship between two agents as follows (F, A, N indicates friendly, adversarial, and neutral respectively):

$$
\begin{aligned}
P(\text{F}|s^{1:T}) = \quad & P(\alpha_{ij} > 0 \text{ or } \alpha_{ji} > 0|s^{1:T}) \\
& + P(g_i = g_j|s^{1:T}) \\
& \cdot P(\alpha_{ij} = 0, \alpha_{ji} = 0|s^{1:T}),
\end{aligned}
\tag{3}
$$

$$
\begin{aligned}
P(\text{A}|s^{1:T}) = \quad & P(\alpha_{ij} < 0 \text{ or } \alpha_{ji} < 0|s^{1:T}) \\
& + P(\text{conflicting } g_i \& g_j|s^{1:T}) \\
& \cdot P(\alpha_{ij} = 0, \alpha_{ji} = 0|s^{1:T}),
\end{aligned}
\tag{4}
$$

and

$$P(\text{N}|s^{1:T}) = 1 - P(\text{F}|s^{1:T}) - P(\text{A}|s^{1:T}), \tag{5}$$

where conflicting $g_i$ and $g_j$ include two types of scenarios: (i) two agents have the goal of putting the same object on different landmarks, and (ii) agent $i$ has the goal of approaching agent $j$ while agent $j$ has the goal of avoiding agent $i$.

This same model can be used to simulate future trajectories based on the goal and relation inference. Specifically, we simulate future trajectories for the most likely hypothesized goals and relationships inferred from the prior observation.

We provide a sketch of SIMPLE in Algorithm 2, where $G(\cdot)$ is our simulation, $P(t_{l,m}|\hat{s}_{l,m}^{1:T}, s^{1:T}, \eta) \propto e^{\eta \sum_{\tau=t_{l,m}}^{t_{l,m}+\Delta T} ||\hat{s}_{l,m}^{\tau} - s^{\tau}||_2}$. For all experiments, we set $L = 6$, $M = 15$, $\eta = 0.1$, $\beta = 0.05$, and $\Delta T = 10$.

### D.1 Bottom-up Proposals

We devise a bottom-up proposal based on heuristics extracted from observed trajectories within a time interval $S^{t_1:t_2}$, i.e., $Y \sim Q(Y|S^{t_1:t_2})$. In this work, the proposal distribution is decomposed into separate terms for proposing goals $(g_i, g_j)$, social utility weights $(\alpha_{ij}, \alpha_{ji})$, and strengths $(f_i, f_j)$ respectively, i.e.,

$$
\begin{aligned}
Q(Y|S^{t_1:t_2}) = \quad & Q(g_i|S^{t_1:t_2})Q(g_j|S^{t_1:t_2}) \\
& \cdot Q(\alpha_{ij}, \alpha_{ji}|S^{t_1:t_2}) \\
& \cdot Q(f_i|S^{t_1:t_2})Q(f_j|S^{t_1:t_2}).
\end{aligned}
\tag{6}
$$

We define the goal proposal distribution for an agent by

$$
\begin{aligned}
Q(g|S^{t_1:t_2}) \quad & \propto e^{\gamma||s_i^{t_2}-s_g||_2} e^{\gamma(||s_i^{t_2}-s_g||_2-|s_i^{t_1}-s_g||_2)} \\
& \propto e^{\gamma(2||s_i^{t_2}-s_g||_2-|s_i^{t_1}-s_g||_2)},
\end{aligned}
\tag{7}
$$

where $\gamma = 10$ is a constant weight. Intuitively, if the trajectories have demonstrated either achievement at the end of the period ($t_2$) or progress towards a goal during the period (from $t_1$ to $t_2$), then that goal is likely to be the true goal. For the social utility weights, we first randomly select $u \in \{-1, 0, 1\}$. If $u = 0$, we set both $\alpha_{ij}$ and $\alpha_{ji}$ to be zero; if $u \in \{-1, 1\}$, we randomly select either $\alpha_{ij}$ or $\alpha_{ji}$, and set it to be $u$ while setting the other one to be zero. This is essentially assuming that there will be at most one agent pursuing a social goal in a social event. For the strengths, we train a 2-layer MLP (64-dim for each layer) using training data in PHASE to estimate the maximum forces that agents can exert.

---

**Algorithm 2:** Sketch of SIMPLE

---

**Input:** $s^{1:T}$, $L$, $M$, $\eta$, $\beta$, $\Delta T$
**Output:** $\{Y_{L,m}\}_{m=1}^{M}$ and their weights $\{w_{L,m}\}_{m=1}^{M}$
**for** $m = 1, \cdots, M$ **do**
     Initial proposal $Y_{0,m} \sim Q(Y_{0,m}|S^{1"T})$;
     Synthesize trajectories $\hat{s}_{l,m}^{1:T} \leftarrow G(Y_{0,m})$;
     $w_{0,m} = \frac{P(s^{1:T}|Y_{0,m})}{\sum_{k=1}^{M} P(s^{1:T}|Y_{0,k})}$;
**end**
**for** $l = 0, \cdots, L-1$ **do**
     **for** $m = 1, \cdots, M$ **do**
         Sample a step $t_{l,m} \sim P(t_{l,m}|\hat{s}_{l,m}^{1:T}, s^{1:T}, \eta)$;
         Set $S' = s^{t_{l,m}:t_{l,m}+\Delta T}$;
         Sample a new proposal $Y' \sim Q(Y_{l+1,m}|S')$;
         Synthesize trajectories $\hat{s}_{l+1,m}^{1:T} \leftarrow G(Y')$;
         $\alpha = \min\{1, \frac{Q(Y'|S')P(s^{1:T}|Y')}{Q(Y_{l,m}|S')P(s^{1:T}|Y_{l,m})}\}$;
         Sample $u \sim \text{Uniform}(0,1)$;
         If $u < \alpha$, $Y_{l+1,m} \leftarrow Y'$, otherwise $Y_{l+1,m} \leftarrow Y_{l,m}$;
         $w_{l+1,m} = \frac{P(s^{1:T}|Y_{l+1,m})}{\sum_{k=1}^{M} P(s^{1:T}|Y_{l+1,k})}$;
     **end**
**end**

---

# E Additional Details of Human Experiments

## E.1 Experiment 1

We recruited 130 participants to label 20% of the videos in PHASE. Each participant was asked to watch a video and select which of the 23 types of interactions was depicted in the video. In total, each video was judged by 10 participants. We found that all 23 types of interactions were selected to describe at least one video. The 23 labels used in this experiment are: not interacting, interacting unintentionally, chasing, running away, stalking, approaching, avoiding, meeting, gathering together, guiding, following the lead (of another agent), playing a game of tag, blocking, fighting, competing, stealing, protecting an object, attacking, hindering, bullying, playing tug of war, helping, collaborating.

## E.2 Experiment 2

Using interface, we collected 8 videos by asking three pairs of human controllers to play with each other in 8 scenarios (covering all three types of relations) that were matched with the scene configurations of 8 videos in PHASE. For the second part, we recruited 15 additional participants.

The online game procedure is similar to the setup in PHASE, except that actions are obtained from user input. In each game, there are two players, one for controlling each agent. The players view the environment from separate screens (via different URLs), updated with each agent's observations. Players can use the following actions by pressing keys on their keyboards: 4 directions (forward, backward, right, left), turning right or left , and grabbing or letting go of an object. We reset the velocity of each agent to 0 after each step to make it easier for players to control the agents. Before each session, the players were shown a tutorial on how the agents work (partial observerability and the controls). They were given an opportunity to play freely in the game environment to get familiar with the controls. At the beginning of a session, they were told the goals assigned to both players (so they knew each others' goals) and asked to start playing the game to achieve the assigned goals. Each session ended either until the goals of both players were achieved or until the time limit was reached.

# F    Baseline Implementation

For all neural nets, we construct the inputs as a sequence of states of multiple nodes. In particular, a node could be an entity, a landmark, or a wall. For a node, the input at a step includes a 4-dim one-hot vector for type (agent, object, landmark, or wall), color (which also indicates the identity of each entity), size, position, orientation, and velocity. We provide implementation details of each baseline as follows.

## F.1    Task 1

**Human:** We collected human judgments of goals and relations on the testing videos from Mechanical Turk. We recruited 130 participants with each being shown 8 videos. After a training phase, participants were instructed to watch videos and choose which of 16 goals each agent had as well as rate the relationship between agents as friendly, neutral, and adversarial.

**2-Level LSTM:** A hierarchical LSTM-based model for recognizing individual actions and the overall group activity. We replace the CNN-based visual features in [10] by node embeddings. Specifically, we encode each node using a 64-dim fully-connected layer followed by an LSTM (64-dim) to get its embedding. For agent nodes, their node embeddings are fed to a 3-layer MLP (64-dim for each layer) and then a softmax layout for goal recognition. After a max pooling over all nodes' embeddings, we get a context feature, which is fed to another LSTM (64-dim) followed by a fully-connect layer and a softmax layer for relation recognition.

**ARG:** Actor Relation Graph, a graph neural net modeling human relations and interactions. We use the same node embedding approach introduced above. Following the best performing architecture in [23], we construct the graph using (i) dot-product for the appearance relation, and (ii) distance mask for position relation (the distance threshold is 8). The individual action classifier for each agent node (3-layer MLP with 64-dim for each layer) and the group activity classier (3-layer MLP with 64-dim for each layer) are redefined to recognize agents' goals and relation respectively.

Both **2-Level LSTM** and **ARG** are trained using a cross-entropy loss on goal and relation labels. We use Adam [13] with a learning rate of 0.001 and a batch size of 8.

## F.2    Task 2

**Social-LSTM:** LSTM-based trajectory prediction with a social pooling mechanism. We adopt the same architecture as in [1] except that for every step, it outputs a 10-step prediction for each entity node. This is to solve our online prediction task.

**STSAT:** Spatial-Temporal Graph Attention network, a state-of-the-art multi-person trajectory prediction approach. We use the same architecture as in [9] for the encoder components. Similarly to the adaption of **Social-LSTM**, the decoder LSTM outputs a 10-step prediction for each entity at each step as well.

We adopt $\mathcal{L}_2$ distance between the prediction and the ground-truth as loss to train the two trajectory prediction baselines. For network optimization, we use Adam with a learning rate of 0.01 and a batch size of 8.

We use two evaluation metrics common in prior work on trajectory prediction [1]: Average Displacement Error (ADE), i.e., average L2 distance between ground truth and the prediction over all steps, and Final Displacement Error (FDE), i.e., the distance between the predicted position and the ground-truth position at the last step. Note that we compute the distance only based on the positions of the entities and do not consider their velocities and angles.

