# OpenReview forum: "PHASE: PHysically-grounded Abstract Social Events for Machine Social Perception"
_NeurIPS.cc/2020/Workshop/SVRHM — SVRHM@NeurIPS Oral_

### Official Review · AnonReviewer2 · 2020-10-29
**Valuable dataset contribution with strong baseline evaluations**

**Rating:** 9
**Confidence:** 3

**Review:**

The simulation for the datasets seem to be designed carefully constructed, diverse in physical and social variables and goals and validated by data from human experiments. The dataset is a valuable and relevant contribution to the workshop.

The baselines (Social LSTM, STGAT) are recent and strong, though they are the “wrong class of models” for this dataset and tasks, which is part of the point, so that SIMPLE is a welcome contribution.

Tables 2, 4: Accuracy -> Error? This makes it confusing, especially since ADE, FDE are only mentioned in abbreviation.

---

### Official Review · AnonReviewer3 · 2020-10-30
**Seemingly outstanding paper: novel dataset and model, rigorously analyzed, although I am not knowledgeable enough to assess the technical details.**

**Rating:** 9
**Confidence:** 2

**Review:**

The paper introduces a new dataset for studying abstract physical-social interaction which, the authors claim (and as far as i can tell it is true) is the first to be physically-grounded. It consists of 2D simple animations of pairs of agents interacting with one another, generated using a physics engine. The authors then conducted 2 human experiments on mechanical turk to validate the dataset. The final dataset consisted of only the animations that were perceived by the participants consistently. Finally, they propose a complex bayesian model for social interaction which outperforms state-of-art-models, at least on the tasks and datasets designed here.

I am not familiar enough with the field of social behavior predictions and social perception to correctly criticize the proposed model or assess the validity of comparing it to the state-of-the-art models in this context. With that said, the paper appears outstanding. The proposed dataset seems rich and complex enough to open up to new studies for understanding social interactions. It has been rigorously validated with human experiments, and can thus be used as benchmark in such studies. The paper is also very well written, the core manuscript easy to follow and the extensive appendices providing the remaining needed technical details.

**Bio Award:**

Yes, paper should be nominated as I have given it a high score and it is also relevant to the award (presents a biologically-driven generative model).

---

### Official Review · AnonReviewer1 · 2020-10-31
**Useful dataset introducing and demonstrating social interactions through geometric shapes**

**Rating:** 9
**Confidence:** 3

**Review:**

This paper presents a very interesting dataset modeling human social interactions using geometric shapes. I particularly found the Experiment 2 verifying the consistency of synthesized trajectories and human-controlled trajectories very insightful.
I believe such a dataset would be useful for a variety of tasks (e.g., training agents for Embodied AI, etc.).

The authors also propose a Bayesian approach for the proposed tasks (SIMPLE). The exposition of this paper can be improved by adding some more details on SIMPLE.

---

### Decision · Program_Chairs · 2020-11-02

Accept (Oral)